# *A guide, cornerstone,* and *appetizer*: An elicited metaphor analysis of Chinese university students' perceptions of English language textbooks

Molly Xie Pan[1], Wei Lin[2], *

**1** Department of English and Communication, The Hong Kong Polytechnic University, Hung Hom, Hong Kong SAR, China, **2** College of Foreign Languages and Literatures, Fudan University, Shanghai, China

* linwei@fudan.edu.cn

## Abstract

Existing research on EFL learners' attitudes towards English language textbooks primarily investigates metaphors at the level of MENTAL SPACES, limiting insights into embodied cognition and experience. This study extends the analysis of metaphors to a more schematic level of DOMAINS/FRAMES. We analyzed 163 metaphors from 123 Chinese university students' perceptions of English language textbooks under the guidance of Conceptual Metaphor Theory and the meta-functions of metaphors in language education. Findings reveal textbooks' three primary roles in learning English as i) a guide in a journey, ii) a cornerstone of a building, and iii) an appetizer in eating. The Chi-Square Test of Independence showed a moderate association between metaphor sources and emotional valence, with NATURE and CONTAINER metaphors associated with negative evaluations. The combination of discourse analysis and statistical analysis highlights learners' physical and emotional engagement with English language textbooks. Pedagogical implications are discussed.

## 1. Introduction

Textbooks assume a pivotal role as the primary carrier of the curriculum [1] and establish an instructional framework that users, including both teachers and students, can navigate [2]. Although the extent to which these textbooks effectively support learning is influenced by their inherent properties, the ultimate impact of textbooks largely hinges on the attitudes of users, with teachers and students collectively shaping how the textbooks are employed inside and outside classrooms [3,4].

The central aim of education is anchored in fostering the comprehensive development of students, recognizing the pivotal role played by participants in the learning process. A prevailing global educational trend emphasizes placing students at the core of the learning experience, a paradigm commonly referred to as student-centered learning. This approach is recognized as a promising strategy for cultivating profound conceptual understanding [5,6]. In the evolving landscape of education, characterized by an increasing emphasis on student-centered approaches, a critical aspect to consider is the examination of students'

**Data availability statement:** All responses from participants are available from Open Science Framework (https://osf.io/hrfqy/?view_only=78bd4a38ee5c4bf492ab096ee9a81f22).

**Funding:** This study is partially supported by Faculty Reserve Grant (P0048130), Faculty of Humanities, The Hong Kong Polytechnic University, Hong Kong SAR, China. There was no additional external funding received for this study.

beliefs and attitudes related to different components of the educational process. Thus, understanding how students perceive and engage with language textbooks and education is crucial for informing and optimizing pedagogical strategies.

The objective of this study is to explore students' perceptions and attitudes towards English textbooks through the lens of metaphors. Within the realm of educational research, metaphors play an important role in providing a "salient, memorable label for an otherwise difficult concept; clarifying a concept that is diffuse, abstract or generally complex; extending thought; or identifying problems with a particular conceptualization and then instigating some form of change" [7]. Metaphoric language is often viewed as links to thought, beliefs, and attitudes [8–12], which can reflect learners' understanding and experience of educational processes. However, existing metaphor studies on learners' perceptions of English language textbooks primarily take place at the level of mental spaces [13–15], i.e., online representations of our comprehension in working memory. This level of metaphor analysis hinders insights into the conventionalized knowledge structures stored in long-term memory, which are vital for comprehending learners' engagement with textbooks.

In this study, we collected 123 surveys on metaphorical perceptions of English language textbooks from EFL learners at a university in China. We identified and analyzed 163 metaphors through both discourse analysis and data analytics techniques. Guided by the Conceptual Metaphor Theory (CMT) [14,16–18], the meta-functions of metaphors in education [19], and Elicited Metaphor Analysis [12,20], we expand the scope of metaphor analysis from online processing in working memory (the level of mental space) to the long-term memory (the level of DOMAINS/FRAMES). The subsequent sections introduce theoretical frameworks and provide detailed methodology. Findings from both discourse analysis and statistical analysis are presented and discussed in relation to existing studies. The paper concludes by summarizing the major findings, discussing practical implications, acknowledging limitations, and suggesting future directions.

## 2. Theoretical background

### 2.1 Definition and levels of metaphor

In the realm of Cognitive Linguistics, a metaphor is defined as "understanding and experiencing one thing in terms of another" [18]. It is a cognitive mechanism that enables us to systematically comprehend and experience a more abstract or unfamiliar concept through a more concrete or familiar one. For instance, in the sentence "when we lose our way in English learning, textbooks **guide** us in the right direction" textbooks are metaphorically portrayed as guides from a rhetorical perspective when a metaphor is deemed as a figurative device to decorate language [21]. However, when a metaphor is regarded as a cognitive mechanism, we can discern that in this context, English learning is depicted as a journey. Learners are understood as travelers, as both can lose their way to their destination and both require the correct direction to follow. The systematic mappings between the domain of English learning and the domain of journey form a conceptual metaphor LEARNING ENGLISH IS A JOURNEY. The domain of English learning is referred to as the target domain, while the domain of journey is the source domain. These mappings are also manifestation of recurring structures in our cognitive processes, known as image schema [17]. The *source-path-goal* schema aids us in understanding and interpreting linguistic meanings in this context. According to the levels of metaphors in Conceptual Metaphor Theory [14], metaphoric language is evident at the level of mental spaces or what Langacker [15] calls "current discourse space". A distinction can be made between the level of mental spaces and the level of systematic mappings (the level of DOMAIN/FRAME) and image schemas. Fig 1 illustrates the levels of metaphors, with image schemas representing the most schematic and mental spaces the least schematic.

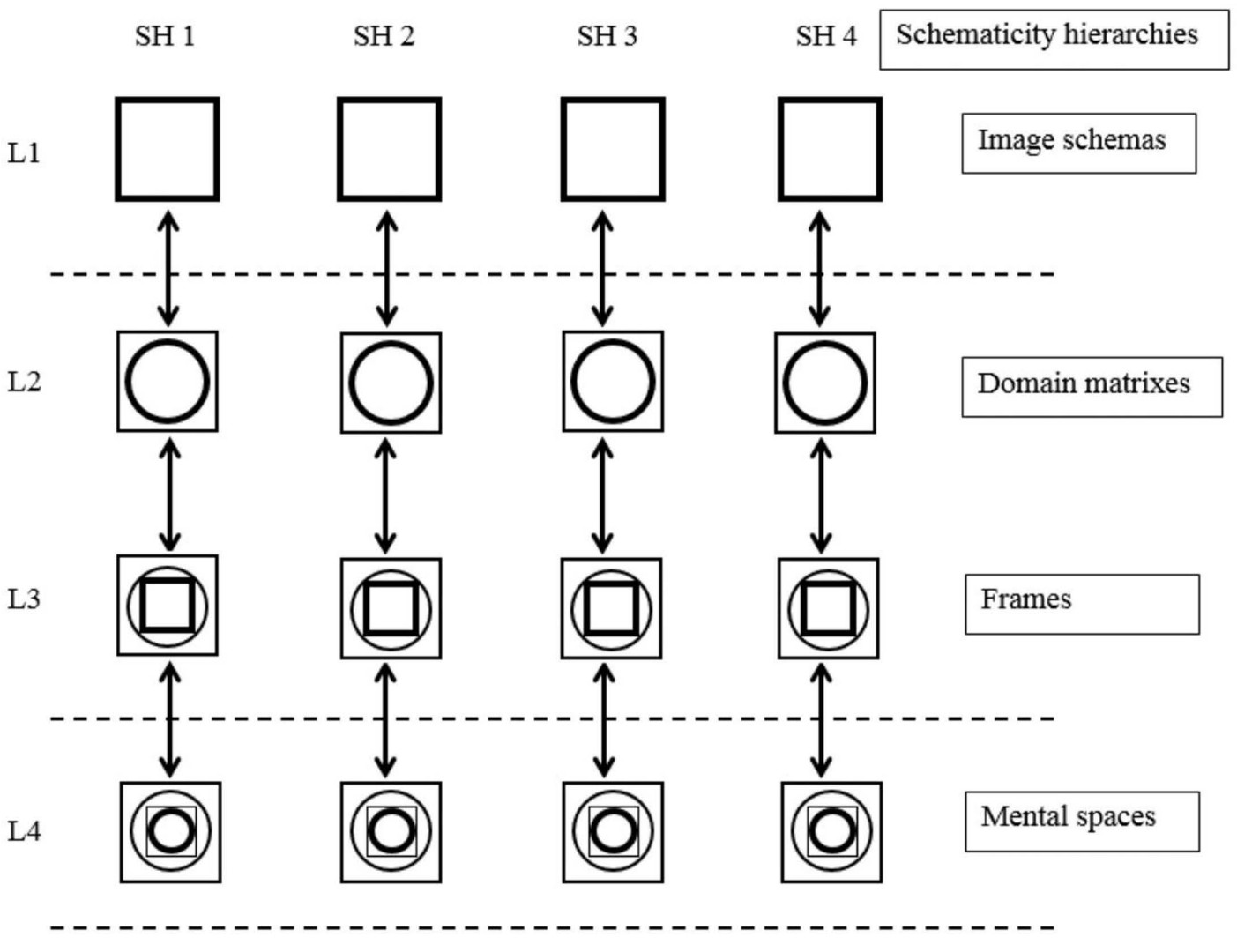

**Fig 1. The schematicity hierarchy with distinctions between levels of metaphor** [14].

## 2.2 Elicited metaphor analysis and textbooks research

Conceptual Metaphor Theory (CMT) has been a source of inspiration for various education-related studies, including those involving metaphor-informed teaching approaches [22–25], metaphoric competence [26,27], and evaluations of textbook content [28,29] to name a few. Researchers also utilize metaphor as a tool to analyze beliefs and attitudes within the learning context [3,9,10,30,31]. Elicited Metaphor Analysis (hereafter, EMA) [12,20,32] is one such method, in which participants are required to provide a metaphor to describe a given topic. In the analysis of EMA, the mapping features from the source to the target are also called entailments. This method is more attuned to metaphorical thinking, enabling us to directly extract insights about the topic [12,20,32].

Researchers have effectively utilized EMA to investigate beliefs and attitudes toward textbooks, encompassing both learners and teachers [3,9,11,33,34]. A predominant approach

in these studies involves categorizing metaphors within a thematic classification framework, notably exemplified by McGrath [3]. Following a bottom-up process, McGrath details steps such as listing and grouping images, identifying semantically related pairs, and establishing working categories like Guidance, Support, Resource, and Constraint. His framework allows flexibility for other possible categories based on data from diverse subject groups.

This categorization approach, often coupled with positive/negative judgments, aids in understanding whether the learning materials meet the specific needs of the learners. It underscores the importance of adapting teaching materials to learner needs and emphasizes the value of surfacing learners' attitudes and their voices. In McGrath's [3] study, a personal collection of images, including both metaphors and similes, representing English language coursebooks is presented. Gathered over two years, this collection includes data from approximately 75 English teachers and several hundred secondary school pupils in Hong Kong. McGrath [3] compares results from teachers and learners, revealing that learner responses cover a broader range, subdivided into more categories and potentially more subcategories than teacher responses. Additionally, while teacher images for coursebooks are predominantly positive with only one negative category (Constraint), learner disaffection encompasses four categories: Constraint, Boredom, Worthlessness, and Anxiety/Fear. This discrepancy highlights varied perspectives between teachers and learners regarding their views on English language coursebooks. Another study by Shi [11], solely focusing on learners, evaluates an English for Academic Purposes (EAP) textbook, examining the beliefs of 147 first-year English as a Foreign Language (EFL) graduate students in a Chinese university. The findings reveal a dual perception of the textbook, providing joy, security, grit, and curiosity, yet being considered old-fashioned, exam-oriented, and teacher-directed. The study by Shi [11] suggests modifications and additional materials to better cater to the needs of EFL graduate students.

### 2.3  Embodied pedagogy: A socio-cultural perspective

While the aforementioned studies generally align with the CMT, their analysis has primarily centered on metaphors at the level of mental spaces (the online processing in the working memory). Mental spaces, as outlined by Kövecses [14], denote cognitive constructs or mental frameworks individuals employ to organize and structure their comprehension of concepts, situations or information. Although all levels of metaphor contribute to the conceptualization [14], analyzing metaphoric language can hardly yield insights into long-term memory without investigating a more schematic level. In the evolving landscape of education where the focus shifts towards valuing individuals and their learning experiences at a more humane level, rather than the mere acquisition of knowledge [35], a comprehensive understanding of how students perceive the education process becomes valuable. This aligns with the principles of embodied pedagogy, urging learners to fully immerse themselves in the learning experience [36].

The analysis of metaphors at a more schematic level can be integrated with a socio-cultural perspective that extends learning from a narrow emphasis on knowledge acquisition and memorization [36,37]. Cortazzi & Jin [19] employ metaphor analysis to scrutinize students' perceptions for "language" in Malaysia. Emphasizing the pivotal role of language and language learning in education, the research identifies patterned language metaphors in proverbs and teacher discourse about literacy. Their study introduces a landscape of meta-functions of metaphors in education, encompassing cognitive, affective, socio-cultural, moral-spiritual, and aesthetic functions. These functions help interpret mappings across domains regarding embodied cognition and socio-cultural experience. To illustrate, consider the metaphor LEARNING IS A JOURNEY: The journey can be described as "giving new knowledge" (cognitive), akin to a "joy ride" (affective), serving as "a connection with others" (socio-cultural), representing a

"stairway to heaven" (moral-spiritual), and being an "indescribable beauty" (aesthetically). Analyzing metaphors at a more schematic level with reference to a socio-cultural perspective highlights the importance of engaging in lived, embodied experiences where emotions and cognition intersect.

Therefore, when it comes to investigating EFL learners' beliefs and attitudes towards English language textbooks, we advance two approaches: 1) extending the metaphor analysis to a more schematic level, and 2) incorporating a socio-cultural perspective on learning experiences. In our study, we will employ EMA and explore the systematic mappings in metaphorical DOMAINS/FRAMES through connecting metaphor analysis at the mental spaces. Our research is guided by the following three research questions:

1   How do learners perceive the role of textbooks in English learning?

2   How do learners' images for textbooks connect with embodied cognition and experience?

3   What are their attitudes towards English textbooks through the use of metaphors?

## 3.  Method

### 3.1  Data collection

The data collection process included online surveys, utilizing convenient sampling. The survey was designed using Elicited Metaphor Analysis [12,20]. Participants were asked to express their perceptions of English language textbooks by completing the sentence "在学习英语的过程中，教材像________，因为__________" [Translation: In the process of learning English, textbooks are (like)_______, because______]. The survey was conducted between September 20, 2023 and December 10, 2023. Data were collected from two English language instructors' classes. At the beginning of the questionnaire, participants were informed in writing that their participation was entirely voluntary. Completing the questionnaire indicates that they have provided consent to participate in this study. Ethics approval was granted by Fudan University Ethics Committee (Approval Number: FE231541). The study did not involve minors, and therefore, no parental or guardian consent was required. Initially, we received 130 completed surveys. The first author reviewed each response, excluding those that did not contain metaphors. For example, *textbooks are (like) reading materials, because we improve our English reading proficiency by reading them*. This response was excluded as the reading materials are part of the content of textbooks. It shows a part-for-whole relationship rather than the cross-domain mappings in metaphors. Ultimately, we collected 123 valid surveys. We conducted word segmentation using *TagAnt* [38], taking into account the character-based nature of Chinese, where a Chinese word can be composed of one or more characters [39,40]. After a manual check and correction, a total of 1,111 tokens were obtained. The respondents were university students from various disciplines. Their English proficiency was determined to be above average based on the college English placement test administered at their university.

### 3.2  Coding of images for textbooks

The classification of images for textbooks, as proposed by McGrath [3], was employed as a guide during the coding of each response with reference to the explanations. However, upon application to the dataset, it was observed that certain categories were absent and some required recategorization, primarily due to cultural differences. For example, in the context of education in Chinese mainland, a teacher is generally viewed as an authority figure. As a result, we coded teacher as "Authority" rather than "Resource". Additionally, some responses

that are difficult to categorize within any specific dimension fall into the 'Ambiguous Image' category.

The coding of image types underwent an inter-rater reliability examination conducted by two raters. They first calibrated their understandings of the scheme by coding 24 responses. Then, they independently coded another 20% responses randomly selected from the dataset. This examination yielded moderate support from Kripendorff's alpha ($k = .79$) [41]. Finally, they resolved disagreements on two cases through discussions.

### 3.3 Metaphor identification

Metaphors were identified from each response, including the explanations. The process of source identification was subjected to rigorous identification procedures. We identified the sources of metaphors by following the Metaphorical Source Domain Identification Procedure (MSDIP) [42]. MSDIP is an extension of the Metaphor Identification Procedure Vrije Universiteit (MIPVU) [43], which addresses the challenge of discerning the source domain from a multitude of potential source domains, with reference to co-text and context [42]. Adhering to a bottom-up approach, we employed MSDIP to identify the sources of metaphors in our data.

Consider this response: *Textbooks are street lamps, because the textual resources provided by the textbooks can guide us to explore a broader path of language learning* [Original Text: 教材/像/路灯，因为教材/提供/的/文本/资源/能/指引/我们/去/探索/更加/广阔/的/语言/道路]. The first step involves a thorough reading of the response to gain a holistic understanding of the context. Next, the contextual meaning of each lexical unit is interpreted. This response highlights the enlightening role of textbooks play in learning English. Subsequently, the basic meaning of the lexical unit is determined by referencing 在线新华字典 [44], or 汉典 [45], following the methods in previous studies [40,46]. If the contextual meaning contrasts with the basic meaning but can be understood in comparison with it, the unit is marked as metaphorical. For instance, a street lamp, by definition, is *a light installed along roads to provide illumination.* In the response, the illuminating function of the lamp is transferred to textbooks. In the elaboration, the units of 探索 *(explore)* and 道路 *(avenue)* are also metaphorical, as they indicate *study* and *language development approaches*, though their definitions denote physical exploration in a journey and avenues. The dictionary definition is then employed to label the potential source domain for each of the more basic meanings. The most likely source-domain candidate is then identified based on co-text and/or context. While our dataset did not present any examples with multiple sources, it is evident that co-text and/or context play a pivotal role in determining a metaphor's source. For example, the definition of a street lamp suggests OBJECT as the source. However, its explanation (the co-text) enables readers to select the illuminating function in a journey as the mapping feature to understand textbooks, signaling the frame of JOURNEY. We, therefore, coded the unit of a street lamp as JOURNEY.

A total of 163 metaphors were identified from the dataset. An inter-rater reliability test was conducted between the first author and a native Chinese speaker who has been engaged in metaphor research for over three years. Initially, they calibrated their understanding of the procedures by examining 10 responses from the dataset. Subsequently, they independently coded 33 randomly chosen responses, which constituted 20.2% of the dataset [47]. The results derived from Krippendorff's alpha indicated an acceptable reliability on the identification of sources ($k = .72$) [41]. The cases of disagreement included responses related to the educational process, but some exhibited distinct categories from textbooks, while others demonstrated a part-for-whole relationship. After discussion, the two raters decided to exclude responses such as "textbooks are exercise books" or "textbooks are reading materials". However, if a response referred to a specific category of book, such as a dictionary, which is part of the educational process, it was retained as a metaphor.

## 3.4 Coding emotional valence

The emotional valence of each metaphor is determined through coding based on the context and the intuition of native speakers. The categories of emotional valence consist of positive, neutral, and negative [46,48,49]. Positive valence refers to responses that only emphasize the benefits and positive assistance from textbooks. For example, the metaphor in the answer 教材/像/路标/因为/帮助/我/更/好/地/学习/英语 [Translation: The textbooks are like a road sign, because it helps me better learn English] is coded as positive due to its focus on the positive aspects. Neutral valence includes responses that exhibit a neutral attitude towards textbooks, as well as those that present both advantages and disadvantages. For instance, the metaphor 教材/像/工具/因为/是/学习/的/工具 [Translation: Textbooks are like tools, because they are tools for learning] is coded as neutral because it does not indicate a specific evaluation of textbooks. Another metaphor, 教材/像/床/因为/我们/需要/床/来/睡觉/但/不/能/一辈子/都/赖/在/床上[Translation: Textbooks are like beds, because we need a bed to sleep, but we cannot lie in bed all our lives], acknowledges the importance of textbooks while also highlighting their limitations. Metaphors in these types of responses are coded as neutral. Negative valence refers to responses that express a negative attitude towards textbooks, such as the metaphor in the answer 教材/像/门/因为/隔绝/了/我/和/英语/1识 [Translation: Textbooks are like a door, separating me from English knowledge]. It implies that textbooks hinder learning the English language.

In accordance with these categories, the emotional valence of each metaphor identified in the dataset was coded. During this process, context (the explanation) was utilized to assist in the coding. For example, while the metaphor of a "door" might conventionally convey a positive emotional valence, the explanation that it "separates me from English knowledge" suggests a negative emotional valence, as it implies that the door acts as an obstacle to accessing English knowledge. Therefore, this metaphor was coded as negative. The authors of this study conducted an inter-rater reliability assessment of coding emotional valence. They first calibrated their understanding of the coding scheme by coding 20 metaphors and then independently coded another 20% of the metaphors randomly selected from the dataset. Disagreements on four cases were resolved through discussions. The raters encountered disagreements when determining whether a metaphor was positive or neutral, particularly when the explanation highlighted both advantages and disadvantages. Take a response as an example: *Textbooks are containers because learning without textbooks is fine, but their absence may lead to chaotic learning.* One rater considered this metaphor positive, as it indicates that textbooks aid in organizing learning. However, the other rater classified it as neutral, given the learner's suggestion that textbooks might not be necessary. After discussion, they decided to categorize such cases as neutral, as the mention of disadvantages reflects a holistic perspective in evaluation. This decision ensures that positive metaphors distinctly convey only positive emotional valence. The results from Krippendorff's alpha indicated a reliable identification ($k = .82$) [41].

## 3.5 Data analysis

The data were analyzed using mixed methods, guided by theoretical frameworks of Conceptual Metaphor Theory (CMT) [14,16–18], Elicited Metaphor Analysis [12,20,30], and the meta-functions of metaphors in language education [19]. Initially, the study extracted prominent conceptual metaphor themes from the dataset, following the framework of CMT and the Master Metaphor List [50]. The functions of textbooks in shaping these conceptualizations were further analyzed with reference to the meta-functions of metaphors in language education, including cognitive, socio-cultural, moral-spiritual, and aesthetic functions [19]. The study then explored the connections between images used for textbooks and the underlying

metaphorical sources, illustrating how images for textbooks at the mental spaces [14] are linked to their conceptualization. Furthermore, the study addressed the third research question by investigating the associations between metaphorical sources and emotional valence as a means of evaluating textbooks, using the Chi-Square Test of Independence. The results of the inferential statistics were interpreted with specific responses. The combination of qualitative and quantitative analyses contributed to the investigation of EFL learners' beliefs and attitudes towards English language textbooks from a Cognitive Linguistics perspective.

## 4. Results

### 4.1 Three conceptual metaphors

There are three conceptual metaphors emerging from the dataset: i) learning English is a journey, ii) learning english is building, and iii) learning english is eating. Together, these metaphors comprise 75 instances, accounting for 46% of the dataset. The remaining metaphors can be considered as ad hoc or novel metaphors, as they do not extend from these Conceptual Metaphors [51,52]. For example, one participant described textbooks as a nice playlist, suggesting that they are interesting and enjoyable to learn.

**4.1.1 Learning English is a journey.** A total of 32 journey metaphors were identified from the dataset. Systematic mappings can be found between the domain of JOURNEY and the domain of LEARNING ENGLISH. The participants described learning English as embarking on a journey, characterized by a definitive goal, a direction to adhere to, and a path to select. In the context of English learning, learners have proficiency objectives to accomplish, language facets to focus on, and learning strategies to employ. The responses depicted this learning process as encompassing activities such as walking, running, exploring, and adventuring in a journey. These activities, therefore, can be interpreted as metaphors for how learners perceive their progression in mastering the English language. Therefore, we could derive a conceptual metaphor LEARNING ENGLISH IS A JOURNEY. Learners may assimilate knowledge at a standard pace (walking) (refer to case 1 in Extract 1) or strive to keep pace with others (running). They may also proactively seek to learn novel concepts and step beyond their comfort zone to acquire new knowledge and/or skills (refer to case 2 in Extract 1). Case 2 in Extract 1 indicates that some learners consciously employ textbooks as a signpost for exploring diverse cultures. This is consistent with the aim of cultivating learner's autonomy during the creation of textbooks [53].

Extract 1

(1) 教材/像/**鞋**/因为/没有/英语/教材/学/英语/**寸步难行**
Translation: Textbooks are like **shoes**, because without English textbooks, it is **hard to take a single step** in learning English.
(2) 教材/像/**路牌**/因为/**涉猎**/各类/文章/各种/文化/**引导**/学生/自主/**深入**/学习
Translation: Textbooks are like **signposts**, because they cover various articles and cultures, **guiding** students to independently delve **deeper** into their studies.
(3) 教材/像/**灯塔**/因为/当/我们/在/英语/学习/中/**迷失**/**方向**/时/教材/为/我们/**指明**/**方向**
Translation: Textbooks are like **lighthouses**, because when we **lose** our **way** in English learning, textbooks **guide** us in the right **direction**.
(4) 教材/像/**灯塔**/因为/**引领**/**方向**/但/不/决定/**航线**
Translation: Textbooks are like **lighthouses**; they **lead the way**, but do not determine **the ship route**.

It was observed that respondents frequently referred to their educational experience as a "voyage" in several instances. Voyage refers to a specific type of long journey by ship. Textbooks are commonly conceptualized as entities that provide guidance throughout this

educational journey. Lighthouses and compasses, both emblematic of navigation during a voyage, are commonly employed. For example, one respondent likened the textbook to a lighthouse, emphasizing its role in providing direction when encountering challenges in learning English (see case 3 in extract 1). Another respondent also described the textbook as a lighthouse, highlighting its function in providing direction without dictating the specific route of the sailing (see case 4 in extract 1). The complexities inherent in a voyage, particularly for individuals residing far from the coast, add a layer of unfamiliarity. Another feature of a voyage is less physical movement compared to hiking, with the use of a vehicle being essential and the inherent risk of maritime accidents. The use of JOURNEY metaphors within the context of a voyage may suggest the concerns and challenges faced by these EFL learners in learning English. Throughout this process, textbooks play a pivotal role in providing guidance and direction. All metaphors in Extract 1 pertain to the socio-cultural aspects of language education [19]. They establish a connection among textbooks, the process of learning English, and the movement between places through embodied experience from the society and culture.

**4.1.2 Learning English is building.** A total of 28 metaphors pertaining to BUILDING have been extracted from the dataset. The process of learning English is depicted as a building project, which necessitates the presence of a blueprint for an overarching plan, a detailed construction design, and the establishment of a solid foundation. Indeed, the learning process encompasses the acquisition of an outline and foundational knowledge, both integral to achieving advanced proficiency. A conceptual metaphor LEARNING ENGLISH IS BUILDING can be established from the responses as an extension of THEORIES ARE BUILDINGS.

Extract 2

(1) 教材/像/**奠基石**/因为/通过/教材/**打**/好/**基础**/**掌握**/基本/英语/技能/未来/可以/更/**轻松**/地/**提升**/英语/能力
Translation: Textbooks are like **foundation stones**, because they **lay the groundwork** for mastering basic English skills, making it easier to enhance English proficiency in the future.

(2) 教材/像/**地基**/因为/课堂/内容/可以/**基于**/课本/内容/拓展
Translation: Textbooks are like a **base**, because classroom content can be expanded **based** on textbook content.

(3) 教材/像/**窗口**/因为/**背后**/**广阔**/的/语言/魅力/和/文化/需要/自发/**探索**
Translation: Textbooks are like **windows**, as the **vast** charm of language and culture **behind** them requires spontaneous **exploration**.

(4) 教材/像/有/**滤镜**/的/**窗户**/因为/人为/编写/**通往**/英语/的/世界
Translation: Textbooks are like **windows** with **filters,** as they are developed by human, **leading** to the world of English.

In this process, textbooks are metaphorically depicted as a building or part of a building, such as cornerstones, bricks, stairs, and windows. These metaphors underscore the foundational role of textbooks in facilitating English language learning. For instance, case 1 in Extract 2 posits that textbooks are crucial to establishing a knowledge base and primary English skills, thereby enabling further enhancement of English proficiency with less effort. Some learners also emphasize that classroom content is elaborated from textbook material (see case 2 in Extract 2). A few learners described textbooks as windows. For instance, one learner suggested that textbooks serve as windows, behind which lies a vast expanse of language and culture that necessitates self-exploration (see case 3 in Extract 2). Another learner proposed that textbooks are akin to filtered windows, crafted by individuals to provide a gateway to the world of English (see case 4 in Extract 2). These metaphors highlight the pivotal role of textbooks in learning English language and the more profound cultural knowledge with self-exploration. The BUILDING metaphors in Extract 2 also demonstrate

how learners utilize their knowledge about construction from the society to explain the role of textbooks in English language learning, thereby demonstrating the socio-cultural meta-functions of metaphors as well.

**4.1.3  Learning English is eating.**  The study identified 15 metaphors with sources of eating-related objects, including soup, food, cooking manuals, coffee, bread, and so forth. The conceptual metaphor LEARNING ENGLISH IS EATING can be an extension of READING IS EATING. The act of eating encompasses tasting and absorbing nutrients from a variety of food types, such as desserts, beverages, and fruits. Similarly, the process of learning English involves engaging in diverse learning activities to reinforce knowledge, including reading textbooks, watching films, listening to songs, and completing exercises. Participant responses drew mappings between the domain of EATING and the domain of LEARNING ENGLISH, highlighting the contribution of textbooks to the development of competence.

Extract 3

(1) 教材/像/**炸鳕鱼**/因为/英语/教材/可以/**提升**/英语/能力/但/如果/没有/**主食**/**薯条**/ (/平时/的/积累) /是/**吃**/不/**饱**/的
Translation: The textbook is like **fried fish**, because while the English textbook can enhance English proficiency, without the **main course** of **chips** (regular accumulation), it is hard to be **full**.

(2) 教材/像/**牛奶**/因为/它/适合/我/的/学习/状况/在/英语/方面/就/像/**青少年**/正在/**成长**/能够/有效/地/**滋养**/我/的/英语/水平
Translation: The textbook is like **milk** because it suits my learning situation (like a **teenager growing** in terms of English), and it can effectively **nourish** my English proficiency.

(3) 教材/像/**饭**/因为/不得不/**吃**
Translation: Textbooks are like **meals**, because we have to **eat**.

Textbooks are often described as food, beverages, or cooking manuals in this dataset. Certain participants equated textbooks to specific types of food or drink, such as milk, dessert, coffee, bread, oranges, or fried fish. For instance, one participant compared textbooks to fried fish, commenting that without chips as main course, achieving fullness is challenging (see case 1 in Extract 3). The participant also explained that the chips could be interpreted as a metaphor for the accumulation of knowledge derived from daily learning. Fish and chips, being a dish originating from Western culture, are not typically Chinese. This metaphor shows that the learner utilizes socio-cultural knowledge to describe textbooks. Another participant depicted textbooks as milk, asserting its suitability for developing language proficiency, akin to providing nutrition for youth development (see case 2 in Extract 3). Some participants also perceive textbooks as general meals. For example, one response stated that textbooks are akin to meals, as we need them (see case 3 in Extract 3). These metaphors emphasize that textbooks provide knowledge of English, thereby fulfilling the cognitive meta-function inherent in language learning.

## 4.2  Connections between metaphors and images for textbooks

The dataset comprised seven categories of images. Table 1 delineates the occurrence of different sources for these image categories. In alignment with the images identified by McGrath [3], the "Resource" category exhibited the most diversity. It was also observed that this category garnered the highest number of images, indicating its broad scope that encapsulates a wide array of sources. This suggests that learners predominantly perceive English language textbooks as a resource to facilitate their English learning. The most recurrent source is the educational process, wherein learners employ a concept from the educational scenario to characterize textbooks. These metaphors predominantly highlight the informative knowledge

**Table 1. An overview of the distribution of sources for each image.**

| Image | Source | Total |
|---|---|---|
| Authority | education process (8), person (1) | 9 |
| Constraint | building (1), nature(1), object (2) | 4 |
| Guidance | building (1), education process (1), journey (20) | 22 |
| Resource | building (6), container (5), eating (15), education process (16), object (1), game (2), instrument (9), journey (8), nature (4), war (2), object (2) | 70 |
| Support | building (11), container (1), space (1) | 13 |
| Worthlessness | object (1) | 1 |
| Ambiguous Image | animal (2), country (1), person (1) | 4 |
| Total | | 123 |

of English, signaling students' recognition of the comprehensiveness of textbooks. For instance, one participant employed the metaphor of encyclopedia, explaining that it encompasses all types of English articles.

Despite having fewer sources, the "Guidance" category exhibits a relatively high number of images, particularly from the JOURNEY source. This could suggest an association between the concepts of guidance and journey within the dataset. It has been observed that textbooks are often conceptualized as maps or signposts within the journey of English language acquisition, as exemplified in Extract 1. Beyond these items, several participants characterize textbooks as a person providing directions on this journey, such as a guide.

The "Authority" category predominantly comprises the EDUCATION PROCESS source. The data reveals that several participants perceive textbooks as teachers who are accessible at any time, thereby facilitating round-the-clock self-learning. For example, one participant characterized textbooks as a teacher for English writing, due to its constant availability. Beyond the education process, one participant viewed the textbook as a foreigner, reasoning that the English language in textbooks is authentic, enabling extensive learning. These metaphors emphasized the social-interaction aspect of learning, which these participants may prioritize and perceive as significant. It also suggests that Chinese learners view teachers and foreigners as authoritative resources of the native language.

Within the "Support" category, the BUILDING source emerged with the highest frequency. The BUILDING source encompasses both the general concept of a building and its constituents. For example, one participant described a textbook as building a house, because its content includes the principles, methods, exemplary architectural cases, macro-design, general theories, construction methods for different parts of a building, and the handling of details.

### 4.3 Evaluation of textbooks through metaphors

A total of 163 metaphors were identified from the dataset. We carried out analysis on top seven sources with a raw frequency of 129 and a cumulative percentage of 79%. Table 2 presents an overview of the frequency of these sources.

A Chi-Square Test of Independence was performed to evaluate the relationship between sources and emotional valence. A significant correlation was found between the two variables, $X^2$ (12, $N = 12$) $= 21.42$, $p = .04$. The effect size of Cramer's V shows a moderate association (Cramer's $V = 0.29$) [54]. A standardized residual with an absolute value exceeding 1.9 is

**Table 2. List of top seven domains.**

| Source domain | Examples of Keywords | Raw Frequency | Valid Percentage | Cumulative Percentage | Total |
|---|---|---|---|---|---|
| Journey | 跑[Run],道路[Avenue] | 32 | 19.63 | 19.63 | 129 |
| Building | 基石[Cornerstone], 地基[Foundation] | 28 | 17.18 | 36.81 | |
| Education Process | 字典[Dictionary], 老师[Teacher], | 24 | 14.72 | 51.53 | |
| Eating | 汤[Soup], 甜点[Dessert] | 15 | 9.20 | 60.73 | |
| Nature | 山[Mountain], 海[Ocean], | 11 | 6.75 | 67.48 | |
| Container | 套子[Case], 收纳箱[Storage Box] | 10 | 6.14 | 73.62 | |
| Body | 头发[Hair], 盲目[Blind] | 9 | 5.52 | 79.14 | |
| Others (10 types) | | 34 | 20.86 | 100 | 34 |
| Total | | | | | 163 |

**Table 3. Cross-tabulation of** SOURCE DOMAIN **and** EMOTIONAL VALENCE.

| Source | | Emotional Valence | | | Total |
|---|---|---|---|---|---|
| | | Negative | Neutral | Positive | |
| body | Count | 0.000 | 1.000 | 8.000 | 9.000 |
| | Standardized residuals | −1.085 | −0.492 | 1.165 | |
| building | Count | 3.000 | 3.000 | 22.000 | 28.000 |
| | Standardized residuals | −0.027 | −1.008 | 0.864 | |
| container | Count | 3.000 | 3.000 | 4.000 | 10.000 |
| | Standardized residuals | 2.027[a] | 1.133 | −2.356[b] | |
| eating | Count | 2.000 | 3.000 | 10.000 | 15.000 |
| | Standardized residuals | 0.329 | 0.323 | −0.498 | |
| education process | Count | 0.000 | 4.000 | 20.000 | 24.000 |
| | Standardized residuals | −1.895 | −0.056 | 1.361 | |
| journey | Count | 2.000 | 5.000 | 25.000 | 32.000 |
| | Standardized residuals | −0.965 | −0.248 | 0.877 | |
| nature | Count | 4.000 | 3.000 | 4.000 | 11.000 |
| | Standardized residuals | 2.844 | 0.942 | −2.762 | |
| Total | Count | 14.000 | 22.000 | 93.000 | 129.000 |

[a]Green cells show that the frequency is more than expected at a statistically significant level.

[b]Red cells indicate that the frequency is significantly less than expected.

regarded as significant [46,55]. Pairs that occur more frequently than expected are highlighted in green cells, while pairs that occur less frequently than expected are in red cells. From Table 3, it can be inferred that CONTAINER metaphors tend to be associated with negative evaluations and avoid positive evaluations at a statistically significant level. Extract 4 showcases two responses that negatively evaluate textbooks through the use of CONTAINER metaphors. In case 1, the learner explicitly stated that the knowledge provided by the textbook is limited, implying that textbooks fall short of meeting their needs. In case 2, the learner criticized most textbooks for failing to cater to individual needs and emphasized the importance of learning authentic English by enhancing the quality of English textbooks. These metaphors pertain to the cognitive meta-function and highlight learners' desire to extend their learning beyond textbooks.

Extract 4

(1) 教材/像/一个/**载体**/因为/提供/的/内容/很/局限
Translation: Textbooks are like a **container**, because the content they provide is very limited.

(2) 教材/像/**套子**/因为/英语/学习/如果/只/在于/课本/并/不/一定/会/激发/学习/兴趣/且/大部分/课本/不/能/**满足**/个性化/的/学习/需求/英语/课本/还/需要/很/大/的/改进/需要/**突破**/**套子**/才/能/学到/真正/实用/的/英语
Translation: Textbooks are like a **case**, because if English learning is only based on textbooks, it may not stimulate interest in learning, and most textbooks cannot **fill** individualized learning needs. English textbooks still need a lot of improvement. We need to **break out** of the **case** to learn authentic and practical English.

Another discernible pattern is the association between NATURE metaphors and negative evaluations. NATURE metaphors are also observed to be less frequent in positive evaluations. Extract 5 presents two responses where NATURE metaphors were employed to negatively evaluate textbooks. In the first case, the learner indicates that the volume of knowledge to be acquired is overwhelming, suggesting a sense of being daunted by the vastness of knowledge. This metaphor delivers both socio-cultural knowledge (the nature of ocean) and a moral-spiritual function (the feeling of being overwhelmed). In Case 2, textbooks are described as mountains, as each article necessitates translation. The term 翻 is polysemous in Chinese, signifying both the act of climbing over mountains and a shortened form of translation. This response employs a metaphorical pun to illustrate the difficulties encountered when using textbooks. This metaphor demonstrates the embodied experience of climbing mountains and cultural knowledge in using translation, and therefore, fulfills the socio-cultural meta-function. While both NATURE metaphors and CONTAINER metaphors are associated with negative evaluations, the contextual analyses reveal divergent learner needs. The CONTAINER metaphors relate to perceptions of the limited knowledge offered by textbooks, urging enhancing the quality of textbooks by using cognitive metaphors. Conversely, the NATURE metaphors fulfill the socio-cultural and moral-spiritual functions. They highlight the challenge of mastering the knowledge contained within textbooks, thereby pointing to the effective utilization of textbooks.

Extract 5

(1) 教材/像/一/片/知识/的/**海洋**/因为/知识点/太/多/了/感觉/一辈子/都/学/不/完
Translation: Textbooks are like an **ocean** of knowledge, because there are so many points to learn that it feels like it would take a lifetime to learn them all.

(2) 教材/像/**山**/因为/每篇/文章/都/要/**慢慢**/**翻**/(翻译/)
Translation: Textbooks are like **mountains**, because each article needs to be **slowly** translated.

It is worth noting that while no obvious tendency is observed for neutral evaluation, the raw frequencies of neutral evaluations and positive evaluations surpass that of negative evaluations. Most learners perceive textbooks as useful and helpful, guiding students on their journey of learning English. Neutral attitudes can also be derived from some responses. Extract 6 shows responses where learners provide a comprehensive view of textbooks, commenting on both their advantages and disadvantages by using knowledge of a building. For instance, in case 1, textbooks are described as cornerstones, but the rationale provided is that they are the foundation, not the whole (edifice) of English. Unlike the cases discussed in Extract 2, where learners only emphasize the fundamental role of textbooks, this response also suggests the limitations of textbooks. In case 2, textbooks are described as desserts or appetizers, followed by an explanation that they function as an introduction or supplement, rather than the primary source for English learning. These explanations for neutral attitudes align with the CONTAINER metaphors, which highlight the limitations of textbooks.

Extract 6

(1) 教材/像/**基石**/因为/它/是/**基础**/但/不是/全部
Translation: Textbooks are like a **cornerstone**, because they provide the **foundation** but not everything.

(2) 教材/像/**甜点**/或/**前菜**/因为/多数/情况/下/发挥/的/是/引入/或/补充/作用/感觉/并非/是/英语/学习/的/主要/**场所**
Translation: Textbooks are like **desserts** or **appetizers**, because in most cases they serve an introductory or supplementary role, and do not seem to be the main **venue** for English learning.

## 5. Discussion

### 5.1 The roles of textbooks in learning English

The three major conceptual metaphors that emerge from the dataset align with previous studies on learners' perceptions of language learning [19,33,56]. These three major conceptual metaphors are i) LEARNING ENGLISH IS A JOURNEY, ii) LEARNING ENGLISH IS BUILDING, and iii) LEARNING ENGLISH IS EATING. Our data show that most learners resort to these conceptual metaphors to elaborate their perceptions of textbooks. In the journey of learning English, textbooks provide the direction for moving on. When English language knowledge is perceived as a building, textbooks serve as the cornerstone underpinning the construction. While some learners regard textbooks as meals, others suggest that textbooks can only be a dessert or appetizer starter, rather than the main course. In summary, our dataset reveals that textbooks primarily serve as guides, supports, and supplements in English learning. Fig 2 provides an overview of the primary functions of textbooks in learning English.

### 5.2 Images and conceptualization of textbooks

Our findings enrich the existing literature by linking images for textbooks with the embodied experience of learning English. Previous studies [3,33] have showed the roles of textbooks as guidance, support, and resources. Our findings further confirm the dominance of these roles and we also found that learners employ their embodied experiences to express their perceptions of textbooks. In our dataset, cognitive and socio-cultural experiences are frequently referenced. However, only a few moral-spiritual metaphors and aesthetic metaphors were found. For instance, textbooks are life-saving straws, because you can cram them (like holding the Buddha's feet) at the end of the semester. The low frequency of such metaphors in our dataset may be attributed to the exploratory nature of our study.

While existing studies were based on the categorization of textbook images [3], we have systematically derived sources for each category. As McGrath [9] pointed out, there are many ways of categorizing images for textbooks. Our approach to the sources is based on established procedures informed by Cognitive Linguistics [17,40,42,43,46]. This approach further refines existing categories and serve as a more specific layer of coding, potentially revealing more structural findings. For instance, analyzing responses at the conceptual level allows us to better understand learners' beliefs and attitudes by uncovering stable structural traits across a large sample size. By incorporating qualitative analysis, our analysis maintains the authenticity of learners' responses, an important advantage of EMA [20]. To summarize, investigating the sources of metaphors at the DOMAIN/FRAME level better serves our research aim of using metaphors as a window into cognition and attitudes, which has also been claimed in previous studies [3,9,11,33].

### 5.3 Evaluation of textbooks

Most learners in our dataset express a positive evaluation, with significantly fewer neutral and negative evaluations, which aligns with previous studies emphasizing positive evaluations as

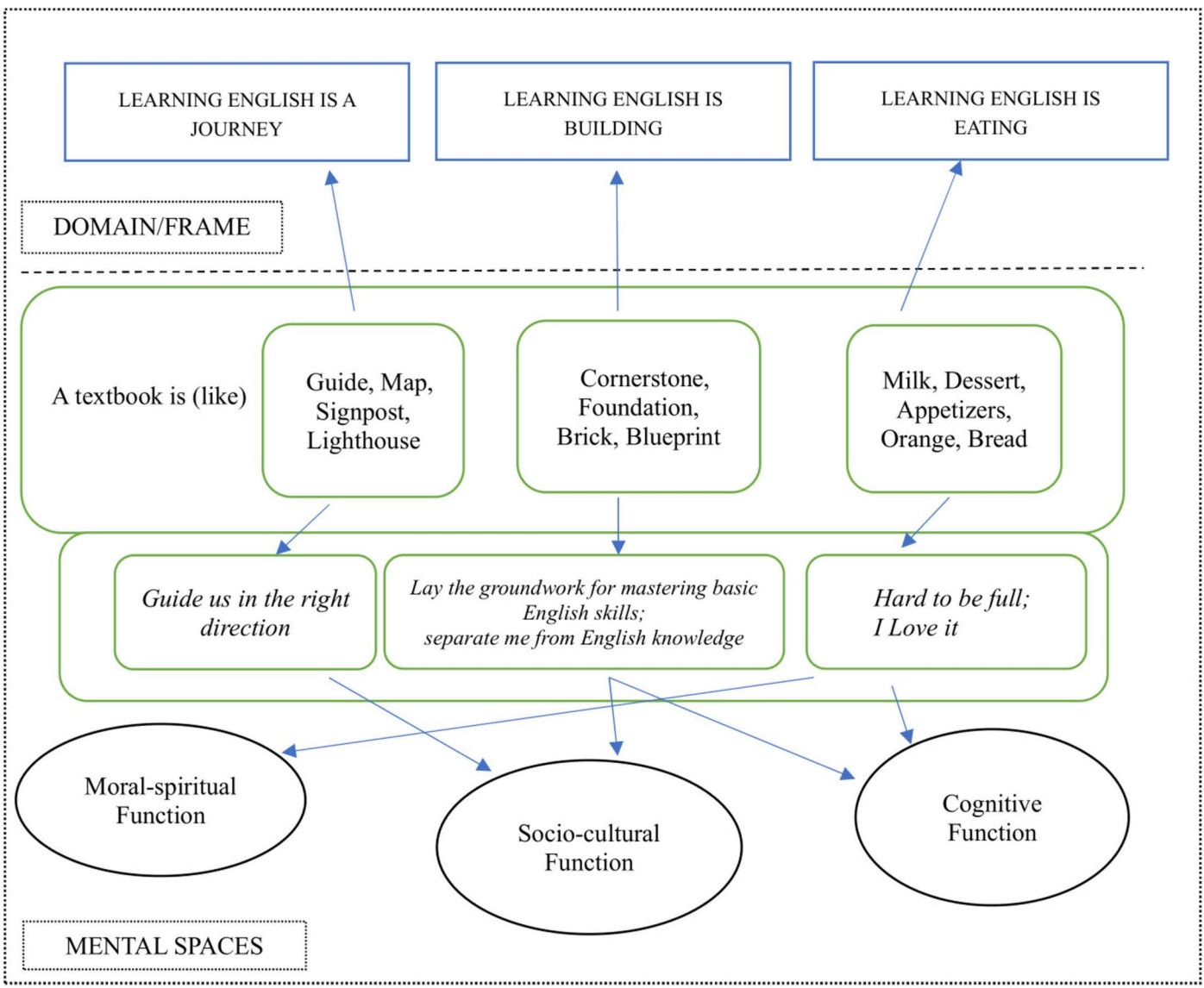

**Fig 2. Primary roles of textbooks in learning English.**

dominant [11,33]. Our study further uncovered that negative evaluations of textbooks are significantly associated with the use of CONTAINER and NATURE metaphors. The strength of the association was moderate, as indicated by the effect size, signaling a moderately stable relationship observable in the target population. Given that metaphor is culturally oriented, understanding its specific use in context is crucial for generating practical implications for textbook design and English learning. We found that some learners employ the CONTAINER metaphor as a resource of embodied cognition to evaluate the knowledge within textbooks. Responses employing this metaphor highlight the limitations of textbooks in fulfilling authentic material needs and individual requirements. This suggests that educators should selectively use the content and provide additional learning materials to broaden the scope of learning. English language learners should also be aware of these limitations and utilize textbooks

according to their specific needs. When developing textbooks for EFL learners, the authenticity of materials should be emphasized [57,58].

There are also participants use NATURE metaphors and refer to socio-cultural knowledge to express their feelings towards the difficulties they encounter while mastering the knowledge in textbooks. Their perceptions reflect the challenge of adopting effective methods for using textbooks in English learning, indicating that they often struggle to find appropriate strategies for acquiring knowledge from textbooks. Some learners appear to be overwhelmed by the task of mastering all the content in the textbooks, while others seem to believe that every article must be translated into Chinese to facilitate their English learning. This finding highlights the need for educators to explicitly teach learners how to use textbooks effectively. Innovative teaching methods, instead of rote memorization of Chinese translations, should be incorporated into English classrooms. Researchers and practitioners must also give greater attention to textbook implementation, as empirical investigations on this topic from learners' perspective remain limited [59]. Effective textbook use at the university level can be supported by incorporating explicit scaffolding into the course design. These efforts may ease learners' transition from high school to university, where textbook use takes on distinct characteristics.

## 6. Conclusion

This study investigates EFL learners' perceptions of English textbooks from a Cognitive Linguistics perspective. We identified 163 metaphors from 123 surveys in which participants responded to the prompt, "English language textbooks are (like)_________, because_______". Research questions focus on the functions of textbooks in the process of English learning, the connections between an established inventory of images for textbooks [3] and the underlying conceptualization, as well as the evaluation of textbooks. Our answer to the first research question is that three major functions of textbooks emerged from major conceptual metaphors in our dataset, i.e., i) LEARNING ENGLISH IS A JOURNEY, ii) LEARNING ENGLISH IS BUILDING, and iii) LEARNING ENGLISH IS EATING. In these three conceptual metaphor themes, textbooks play the roles of guiding, supporting, and supplementing, respectively. In response to the second research question, we found that an image category may employ different sources to deliver the functions of textbooks. In our dataset, socio-cultural metaphors with sources of EDUCATION PROCESS, JOURNEY, and BUILDING frequently appear in images of authority, guidance, and support. More cognitive metaphors with sources of EATING, INSTRUMENT, and OBJECT occur in images of resource guidance, and constraint. A few moral-spiritual metaphors with source of WAR, RELIGION, and BUILDING appear in the image of resource. Regarding the third research question, we found that most learners positively evaluate the assistance role of English textbooks, with significant associations between sources and emotional valence. The Chi-square Test of Independence showed associations between CONTAINER/NATURE metaphors and negative evaluations. Learners use CONTAINER metaphors to evaluate the cognitive aspect of knowledge in textbooks, whereas some learners employ NATURE metaphors which fulfill the socio-cultural and moral-spiritual meta-functions to express challenges in effectively using textbooks.

Our study contributes to the existing literature in the following aspects. First, it addresses the theoretical need to understand learners' beliefs and attitudes towards textbooks by using metaphors as a window into embodied cognition from a Cognitive Linguistics perspective. By integrating Conceptual Metaphor Theory [16,17] and the meta-functions of metaphors in language education [19], we connect the images for textbooks at the mental spaces to a more schematic level of conceptualization. By incorporating an embodied experience perspective, this approach enriches language education research and facilitates the application of Cognitive

Linguistics to the field [35,46]. Second, the mixed-method investigation expands on the qualitative-oriented Elicited Metaphor Analysis [12,20]. The quantitative insights complement the qualitative analysis and guide subsequent analysis within the context. Third, our findings on evaluations offer pedagogical contributions to textbook design and use in EFL contexts. Textbook developers should prioritize the authenticity of materials, while language instructors should address learners' needs by adapting textbook use and incorporating additional authentic learning resources. Emphasizing strategies for effective textbook use in university settings can also help learners adjust to the demands of higher education.

However, our study has three limitations. First, the limited sample size restricts comparisons between students in different disciplines, which could potentially reveal perceptions of discipline-specific textbooks. Furthermore, perceptions of textbooks among some first-year undergraduate participants may evolve as they advance through university. To investigate these diachronic changes, future research may adopt a longitudinal approach. Third, the elicited metaphor analysis method could be improved by providing more instruction on metaphors before the survey. Nacey and Turunen [32] showed that such intervention help elicit more detailed responses and less non-metaphor responses. Further studies could consider the updated elicitation model.

## Acknowledgements

We would like to thank Dr Shanshan Yang for data collection and data coding.

During the preparation of this work the authors used ChatGPT 4.0 to improve language and readability. After using this tool, the authors reviewed and edited the content as needed and take full responsibility for the content of the publication.

## Author contributions

**Conceptualization:** Molly Xie Pan, Wei Lin.

**Data curation:** Molly Xie Pan, Wei Lin.

**Funding acquisition:** Molly Xie Pan.

**Investigation:** Molly Xie Pan, Wei Lin.

**Methodology:** Molly Xie Pan, Wei Lin.

**Project administration:** Molly Xie Pan.

**Writing – original draft:** Molly Xie Pan, Wei Lin.

**Writing – review & editing:** Molly Xie Pan, Wei Lin.

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
