## [Decision Letter · Decision Letter 0]

24 Sep 2024

PONE-D-24-34991A guide, cornerstone, and appetizer: An elicited metaphor analysis of Chinese university students’ perceptions of English language textbooksPLOS ONE

Dear Dr. Lin,

Thank you for submitting your manuscript to PLOS ONE. After careful consideration, we feel that it has merit but does not fully meet PLOS ONE’s publication criteria as it currently stands. Therefore, we invite you to submit a revised version of the manuscript that addresses the points raised during the review process.

**Additionally, we kindly ask you to consider the following PLOS ONE criteria regarding qualitative and mixed-methods studies for publication. Your study should include the following information: "description of the sampling strategy, including rationale for the recruitment method, participant inclusion/exclusion criteria and the number of participants recruited". Please check whether you have taken all those information into account or add any missing information.**

We look forward to receiving your revised manuscript.

Kind regards,

Angelika Pahl

Academic Editor

PLOS ONE

**Journal Requirements:**

This study is partially supported by Faculty Reserve Grant (P0048130), Faculty of Humanities, The Hong Kong Polytechnic University, Hong Kong SAR, China.  

Reviewers' comments:

Reviewer's Responses to Questions

**Comments to the Author**

1. Is the manuscript technically sound, and do the data support the conclusions?

Reviewer #1: Yes

Reviewer #2: Yes

2. Has the statistical analysis been performed appropriately and rigorously? 

Reviewer #1: Yes

Reviewer #2: Yes

3. Have the authors made all data underlying the findings in their manuscript fully available?

Reviewer #1: Yes

Reviewer #2: Yes

4. Is the manuscript presented in an intelligible fashion and written in standard English?

Reviewer #1: Yes

Reviewer #2: Yes

5. Review Comments to the Author

**Reviewer #1:** The researchers take a cognitive linguistic approach to exploring how students who are learning English as a foreign language conceptualize the role of textbooks in this process. They used a qualitative approach known as Elicited Metaphor Analysis that serves to encourage participants to generate metaphors. The researchers emphasize that while previous research has focused on analysis of metaphors from the perspective of mental spaces, which focuses on what is happening in working memory, they want to focus on the perspective of domains/frames, which focuses on what exists in long-term memory. The researchers coded the metaphors generated by students in three different ways. They coded the metaphors in terms of embodied images (e.g., cognitive, affective, socio-cultural, moral-spiritual, aesthetically), metaphorical sources (e.g., JOURNEY, BUILDING, EATING), and emotional valence (e.g., positive, neutral, negative). The researchers found that the students most frequently generated metaphors based on the following conceptual metaphors/sources: LEARNING ENGLISH IS A JOURNEY, LEARNING ENGLISH IS BUILDING, and LEARNING ENGLISH IS EATING. The researchers also found that the responses connected to each conceptual metaphor/source were distributed across a variety of embodied image categories that indicate the primary function of textbooks; however, there appeared to be strong correspondence between JOURNEY and guidance, BUILDING and support, and EATING and (supplemental) resource. These results are consistent with previous literature but add to the literature. While most students had a positive view of textbooks, those generating metaphors based on CONTAINER and NATURE sources tended to have a more negative view of textbooks. The researchers argue that this information can be used by textbook publishers to create more effective products by incorporating authentic materials and addressing challenges that students might encounter.

I saw no major issues with the manuscript. The minor issues that I encountered are as follows. (1) On page 35 the researchers describe their study as a “mixed-method investigation,” but I am unclear on what part of the research design is quantitative. It might be that the researchers are referring to the quantitative analysis that they employed. In this case, my recommendation would be to specifically identify their analysis as mixed-methods. (2) On page 11 the researchers mention excluding responses that did not contain metaphors. I thought it would have been helpful to provide an example. (3) On page 9 the researchers write, “The study suggests modifications and additional materials to better cater to the needs of EFL graduate students.” I am not sure whether “the study” being referred to is the current study or Shi’s (2022) study. (4) When looking at Table 3 on its own, I did not know what the red and green highlights meant. While this is explained in the body of the document, a note could be added to the table for the purposes of clarification. (5) In the reference section, there are some minor APA Style formatting errors. The most consistent one was not including DOI links, which would be easy to resolve. (6) One grammatical issue I noted was the occasional use of commas when not needed (see selected examples below). (7) Another grammatical issue I noted was the occasional awkward wording, which caused me to experience issues with comprehension (see selected examples below).

While portions of the document seem a bit repetitive, what the researchers excelled at was providing good examples to support their arguments.

(3) Unnecessary comma examples

Page 6 - “when we lose our way in English learning, textbooks guide us in the right direction”, textbooks are metaphorically portrayed as guides

Page 7 – and evaluations of textbook content (Deignan, 2017; Deignan & Semino, 2019), to name a few.

Page 17 - Learners may assimilate knowledge at a standard pace (walking) (refer to case 1 in Extract 1), or strive to keep pace with others (running).

(3) Awkward wording examples

Abstract - The synergy of discourse analysis and statistical analysis suggests learners’ physical and emotional engagement with English language textbooks.

Page 12 - a few responses which can hardly be categorized from any dimension

Page 8 - learner disaffection spans four categories

Page 16 - total of 75 metaphors constitute to these Conceptual Metaphors

**Reviewer #2:** The study employs a sound methodology using Conceptual Metaphor Theory (CMT) to analyze Chinese EFL students' metaphors for English textbooks. However, while the data is intriguing, the manuscript lacks deeper theoretical engagement with recent metaphor research. Updating and expanding the literature review to include more recent works would strengthen the paper's relevance. The elicited metaphor analysis (EMA) is well executed, but the coding process for metaphors and emotional valence needs more detail. While Krippendorff's alpha scores indicate reliability, additional transparency on the disagreements during coding would be beneficial.

The study makes an original contribution by expanding metaphor analysis to domains/frames rather than focusing solely on mental spaces. However, the conclusions could benefit from further linking the findings to pedagogical applications or practical suggestions for improving textbook design and use. The metaphorical insights are well connected to the learners' attitudes, but the pedagogical implications need further exploration. The statistical analysis (Chi-Square Test of Independence) is appropriate for examining the relationship between metaphor sources and emotional valence. However, a discussion of the effect size could provide better insight into the practical significance of the findings.

While the discussion mentions implications for EFL learning and textbook design, it would benefit from a more robust exploration of how these metaphorical insights could be applied by educators.

The ethics statement is clear, and the informed consent process is appropriate.

In conclusion, this manuscript offers a promising exploration of metaphorical perceptions in EFL learners, but could benefit from further engagement with recent research and more in-depth discussion of the practical applications of the findings.

6. PLOS authors have the option to publish the peer review history of their article (what does this mean?). If published, this will include your full peer review and any attached files.

Reviewer #1: No

Reviewer #2: **Yes: **Marta Silvera-Roig

---

## [Author Response · Author response to Decision Letter 0]

28 Oct 2024

Dear Editor and reviewers,

We sincerely thank you and reviewers for the valuable comments, which have greatly helped us improve our manuscript. We have extensively revised the manuscript based on the feedback received. All changes are marked by Track Changes and highlighted for easier reading. Below are our responses to comments:

Comments from the editor:

1. Additionally, we kindly ask you to consider the following PLOS ONE criteria regarding qualitative and mixed-methods studies for publication. Your study should include the following information: "description of the sampling strategy, including rationale for the recruitment method, participant inclusion/exclusion criteria and the number of

participants recruited". Please check whether you have taken all those information into account or add any missing information.

Our response: Thank you for reminding us of this issue. We have double checked our manuscript and we hereby confirm that these aspects have been reported in section 3.1 (highlighted in the revised manuscript).

2. Thank you for stating in your Funding Statement: This study is partially supported by Faculty Reserve Grant (P0048130), Faculty of Humanities, The Hong Kong Polytechnic

University, Hong Kong SAR, China.

Please provide an amended statement that declares *all* the funding or sources of support (whether external or internal to your organization) received during this study, as detailed online in our guide for authors at http://journals.plos.org/plosone/s/submit-now.

Please also include the statement “There was no additional external funding received for this study.” in your updated Funding Statement. Please include your amended Funding Statement within your cover letter. We will change the online submission form on your behalf.

Our response: We have included the following amended statement within our new cover letter.

Updated statement: This study is supported by a Faculty Reserve Grant (P0048130), Faculty of Humanities at The Hong Kong Polytechnic University, a Start-up Fund for RAPs under the Strategic Hiring Scheme (P0048980) at The Hong Kong Polytechnic University, and a Pujiang Talent Program (21PJC014) awarded to the first author. There was no additional external funding received for this study.

Our response: We have updated this information in section 3.1 (see pages 10-11).

4. Please include captions for your Supporting Information files at the end of your manuscript, and update any in-text citations to match accordingly. Please see our Supporting Information guidelines for more information: http://journals.plos.org/plosone/s/supportinginformation.

Our response: We have included the captions for the two supporting information files at the end of the updated manuscript.

Our response: Thank you for the reminder. We have checked the reference list to make sure it is complete and correct. None of the cited papers have been retracted.

Comments from reviewer 1:

(1) On page 35 the researchers describe their study as a “mixed-method investigation,”

but I am unclear on what part of the research design is quantitative. It might be that the researchers are referring to the quantitative analysis that they employed. In this case, my recommendation would be to specifically identify their analysis as mixed-methods.

Our response: Thank you very much for pointing out this problem. We have revised the data analysis part in the method (3.5) to explicitly demonstrate that our analysis employs mixed methods. Please see the revised sentences on page 16.

(2) On page 11 the researchers mention excluding responses that did not contain metaphors. I thought it would have been helpful to provide an example.

Our response: Thank you for this suggestion! We have provided an example following this statement. Please see page 11.

(3) On page 9 the researchers write, “The study suggests modifications and additional materials to better cater to the needs of EFL graduate students.” I am not sure whether “the study” being referred to is the current study or Shi’s (2022) study.

Our response: Thank you for pointing out this problem. We have clarified it by adding Shi (2022) into the sentence. Please see page 8.

(4) When looking at Table 3 on its own, I did not know what the red and green highlights

meant. While this is explained in the body of the document, a note could be added to the table for the purposes of clarification.

Our response: Thank you for this concrete suggestion. We have added notes for Table 3. Please see page 26.

(5) In the reference section, there are some minor APA Style formatting errors.

 The most consistent one was not including DOI links, which would be

easy to resolve.

Our response: Thank you for pointing out this problem. We have changed the format to Vancouver style which is required by the journal. We have also added DOI links to references for all those that have a DOI.

(6) One grammatical issue I noted was the occasional use of commas when not needed (see selected examples below).

Unnecessary comma examples

Page 6 - “when we lose our way in English learning, textbooks guide

us in the right direction”, textbooks are metaphorically portrayed as

guides

Page 7 – and evaluations of textbook content (Deignan, 2017;

Deignan & Semino, 2019), to name a few.

Page 17 - Learners may assimilate knowledge at a standard pace

(walking) (refer to case 1 in Extract 1), or strive to keep pace with

others (running).

Our response: Thank you for listing these examples in details. We have addressed these problems one by one. Please see pages 6, 7 and 18.

(7) Another grammatical issue I noted was the occasional awkward

 wording, which caused me to experience issues with comprehension

(see selected examples below).

Awkward wording examples

Abstract - The synergy of discourse analysis and statistical analysis

suggests learners’ physical and emotional engagement with English

language textbooks.

Page 12 - a few responses which can hardly be categorized from any

dimension

Page 8 - learner disaffection spans four categories

Page 16 - total of 75 metaphors constitute to these Conceptual

Metaphors

Our response: Thank you for listing these examples in details. We have rephrased these sentences one by one. Please see pages 3, 11 and 17.

Comments from reviewer 2:

(1) The study employs a sound methodology using Conceptual Metaphor Theory (CMT) to analyze Chinese EFL students' metaphors for English textbooks. However, while the data is intriguing, the manuscript lacks deeper theoretical engagement with recent metaphor research. Updating and expanding the literature review to include more recent works would strengthen the paper's relevance.

Our response: Thank you for this suggestion. We have updated and expanded the literature review with more recent works to address this issue. Please see our changes highlighted in introduction and theoretical background (pages 4 to 10)

(2) The elicited metaphor analysis (EMA) is well executed, but the coding process for metaphors and emotional valence needs more detail.

Our response: Thank you very much for these concrete suggestions! We have added examples to illustrate the coding process of metaphors. Please see the updated sections 3.3 on pages 12-14. We have elaborated the coding process for emotional valence and added an example. Please see the updated 3.4 on pages 14-16.

(3) While Krippendorff's alpha scores indicate reliability, additional transparency on the disagreements during coding would be beneficial.

Our response: Thank you very much for this suggestion. We have added disagreed examples to improve the transparency. Please see the updated sections 3.3 and 3.4 on pages 14-16.

(4) The study makes an original contribution by expanding metaphor analysis to domains/frames rather than focusing solely on mental spaces. However, the conclusions could benefit from further linking the findings to pedagogical applications or practical suggestions for improving textbook design and use.

Our response: Thank you for this suggestion. We have elaborated our pedagogical applications in the conclusion. Please see our revised section on pages 34-35.

(5) The statistical analysis (Chi-Square Test of Independence) is appropriate for examining the relationship between metaphor sources and emotional valence.

 However, a discussion of the effect size could provide better insight

into the practical significance of the findings.

 Our response: Thank you for this comment! We have integrated the discussion of effect size into discussing practical implications. Please see our revised section 5.3 on pages 31-32.

(6) The metaphorical insights are well connected to the learners' attitudes, but the pedagogical implications need further exploration. While the discussion mentions implications for EFL learning and textbook design, it would benefit from a more robust exploration of how these metaphorical insights could be applied by educators.

 Our response: Thank you for this comment! We have elaborated the discussion of practical implications for educators and language learners. Please see our revised section 5.3 on pages 31-33.

(7) The ethics statement is clear, and the informed consent process is

appropriate.

Many thanks for this comment!

(8) In conclusion, this manuscript offers a promising exploration of metaphorical perceptions in EFL learners, but could benefit from further engagement with recent research and more in-depth discussion of the practical applications of the findings.

Our response: Thank you for your concrete suggestions. We have included more recent research and elaborated discussion of the practical applications.

We found that these insightful suggestions significantly enhance our manuscript. We hope our improved version could meet the requirements of this journal. Thank you again for your valuable time in reviewing our manuscript!

Best wishes,

Authors

---

## [Decision Letter · Decision Letter 1]

25 Nov 2024

A guide, cornerstone, and appetizer: An elicited metaphor analysis of Chinese university students’ perceptions of English language textbooks

PONE-D-24-34991R1

Dear Dr. Lin,

We’re pleased to inform you that your manuscript has been judged scientifically suitable for publication and will be formally accepted for publication once it meets all outstanding technical requirements.

Kind regards,

Angelika Pahl

Academic Editor

PLOS ONE

Additional Editor Comments (optional):

Reviewers' comments:

Reviewer's Responses to Questions

**Comments to the Author**

1. If the authors have adequately addressed your comments raised in a previous round of review and you feel that this manuscript is now acceptable for publication, you may indicate that here to bypass the “Comments to the Author” section, enter your conflict of interest statement in the “Confidential to Editor” section, and submit your "Accept" recommendation.

Reviewer #1: All comments have been addressed

Reviewer #2: All comments have been addressed

2. Is the manuscript technically sound, and do the data support the conclusions?

Reviewer #1: (No Response)

Reviewer #2: Partly

3. Has the statistical analysis been performed appropriately and rigorously? 

Reviewer #1: (No Response)

Reviewer #2: Yes

4. Have the authors made all data underlying the findings in their manuscript fully available?

Reviewer #1: (No Response)

Reviewer #2: Yes

5. Is the manuscript presented in an intelligible fashion and written in standard English?

Reviewer #1: (No Response)

Reviewer #2: Yes

6. Review Comments to the Author

Reviewer #1: (No Response)

Reviewer #2: (No Response)

7. PLOS authors have the option to publish the peer review history of their article (what does this mean?). If published, this will include your full peer review and any attached files.

Reviewer #1: No

Reviewer #2: **Yes: **Marta Silvera-Roig

---

## [Editor Report · Acceptance letter]

PONE-D-24-34991R1

PLOS ONE

Dear Dr. Lin,

I'm pleased to inform you that your manuscript has been deemed suitable for publication in PLOS ONE. Congratulations! Your manuscript is now being handed over to our production team.

Kind regards,

on behalf of

Dr. Angelika Pahl

Academic Editor

PLOS ONE